

# An updated evolutionary study of the Notch family reveals a new ancient origin and novel invariable motifs as potential pharmacological targets

Dimitrios Vlachakis[1,2,3], Louis Papageorgiou[4], Ariadne Papadaki[1], Maria Georga[1], Sofia Kossida[5] and Elias Eliopoulos[1]

[1] Laboratory of Genetics, Department of Biotechnology, School of Applied Biology and Biotechnology, Agricultural University of Athens, Athens, Greece
[2] University Research Institute of Maternal and Child Health & Precision Medicine, and UNESCO Chair on Adolescent Health Care, "Aghia Sophia" Children's Hospital, National and Kapodistrian University of Athens, Athens, Greece
[3] Division of Endocrinology and Metabolism, Center of Clinical, Experimental Surgery and Translational Research, Biomedical Research Foundation of the Academy of Athens, Athens, Greece
[4] Department of Informatics and Telecommunications, National and Kapodistrian University of Athens, Athens, Greece
[5] IMGT, The International ImMunoGeneTics Information System, Université de Montpellier, Laboratoire d'ImmunoGénétique Moléculaire and Institut de Génétique Humaine, University of Montpellier, Montpellier, France

Corresponding author
Dimitrios Vlachakis, dimvl@icloud.com

## ABSTRACT

Notch family proteins play a key role in a variety of developmental processes by controlling cell fate decisions and operating in a great number of biological processes in several organ systems, such as hematopoiesis, somatogenesis, vasculogenesis, neurogenesis and homeostasis. The Notch signaling pathway is crucial for the majority of developmental programs and regulates multiple pathogenic processes. Notch family receptors' activation has been largely related to its multiple effects in sustaining oncogenesis. The Notch signaling pathway constitutes an ancient and conserved mechanism for cell to cell communication. Much of what is known about Notch family proteins function comes from studies done in *Caenorhabditis Elegans* and *Drosophila Melanogaster*. Although, *human* Notch homologs had also been identified, the molecular mechanisms which modulate the Notch signaling pathway remained substantially unknown. In this study, an updated evolutionary analysis of the Notch family members among 603 different organisms of all kingdoms, from *bacteria* to *humans*, was performed in order to discover key regions that have been conserved throughout evolution and play a major role in the Notch signaling pathway. The major goal of this study is the presentation of a novel updated phylogenetic tree for the Notch family as a reliable phylogeny "map", in order to correlate information of the closely related members and identify new possible pharmacological targets that can be used in pathogenic cases, including cancer.

## INTRODUCTION

The Notch gene was originally discovered by *Dexter (1914)* and it was named from the irregular notched wing phenotype of *Drosophila melanogaster*, caused by the loss-of-function in the responsible gene's after a point mutation. Since then, Notch protein and its homologs, Notch1, Notch2, Notch3, Notch4, LIN-12 and GPL-1 have been identified in genomes from all kingdoms, indicating the progressive differentiation of Notch family. Their length varies from ≈110 amino acids in *bacteria* (*Durieux et al., 2019*) to ≈4,500 amino acids (aa) in animals (*Fairclough et al., 2013*). Notch family members are evolutionary conserved, type-1 transmembrane glycoproteins, that function both as transmembrane receptors for ligands and transcription factors (*Kopan & Ilagan, 2009*). They regulate cell fate determination and promote cell differentiation, maintenance and survival. These proteins have either overlapping or unique cellular functions, but these functions remain quite unclarified, in the majority of the organisms found (*Hogan & Bautch, 2004*).

In *mammals*, there are four genes that encode four paralogue Notch transmembrane receptors, Notch1 to 4. The Notch1 gene is essential for developmental processes, while loss-of-function mutations in result to early fetal death, due to dysfunctional angiogenesis, organogenesis and cardiogenesis. Moreover, it plays a key role in the definitive formation of Hematopoietic Stem Cells (HSCs), responsible for the production of all mature blood cells during adulthood (*Tanigaki & Honjo, 2007*). Notch1 protein consists of approximately 2,627 aa and its signaling pathway regulates the development of B and T lymphocytes (*Gerhardt et al., 2014*). The Notch2 gene encodes a ≈2,471 aa receptor and determines the cell fate in the heart, liver, kidneys, teeth, bones as well as other cell types that are being developed in the fetus. After birth, Notch2 signaling is involved in the immune system's function, tissue repair, homeostasis and bone reshaping when needed. The Notch3 gene encodes a ≈2,321 aa receptor, which determines the fate of Vascular Smooth Muscle Cells in the arterial network of the brain and finally the Notch4 gene encodes a ≈2,059 aa receptor, that determines fetal vascular morphogenesis and remodeling (*Krebs et al., 2000*; *Vlachakis et al., 2014*).

Notch family protein domains have been conserved throughout evolutionary history, from *invertebrates* to *humans* (*Hori, Sen & Artavanis-Tsakonas, 2013*). They are composed of an extracellular domain (Notch extracellular domain, NECD), a transmembrane domain and an intracellular domain (Notch intracellular domain, NICD) (*Yavropoulou, Maladaki & Yovos, 2015*). The NECD contains 29 to 36 Epidermal Growth Factor-like Repeats (EGF-like domain), depending on the type of receptor and a Negative Regulatory Region (NRR). The NRR is composed of three cysteine-rich Notch/LIN-12 repeats (LNRs) and a Heterodimerization Domain (HD). Each EGF-like repeat has six cysteines, which form three disulfide bonds, contributing to the 3D structure of the protein (*Muiño et al., 2017*). The NICD has a RBPJκ-associated molecule domain (RAM), nuclear localization sequences (NLS), seven ankyrin repeats (ANK) domain, a transcriptional activation domain (TAD) and a C-terminal Pro Glu Ser Thr (PEST) domain (*Hori, Sen & Artavanis-Tsakonas, 2013*). Although Notch receptors are highly conserved, they have some

**Table 1 Diseases which are caused by NOTCH1-4 mutations. The research was done using the DisGeNET database (www.disgenet.org).**

| Diseases | NOTCH Proteins | | | |
| --- | --- | --- | --- | --- |
| | NOTCH1 | NOTCH2 | NOTCH3 | NOTCH4 |
| | • T-cell acute lymphoblastic leukemia <br> • Adams-Oliver Syndrome <br> • Aortic Valve Disease <br> • Cancer | • Hajdu-Cheney Syndrome <br> • Alagille Syndrome <br> • Cancer | • CADASIL <br> • Infantile Myofibromatosis <br> • Early-onset arteriopathy with cavitating leukodystrophy <br> • Lateral meningocele Syndrome <br> • Cancer | Unclarified |

structural variation mainly in the number of EGF-like repeats, in the presence of the TAD domain, and the length of the segment between the ANK repeats and C-terminal (*Sander, Krysinska & Powell, 2006*). The Notch signaling pathway is highly conserved, with a key role in cell-cell communication. It regulates the vascular development and physiology, as well as multiple developmental processes of the Central Nervous System (CNS). The pathway is involved in the regulation of Nerve Stem Cells' (NCSs) proliferation, survival, self-renewal and differentiation and has been associated with early neurodevelopment, learning and memory, as well as neurodegeneration (*Mizutani et al., 2007*; *Polychronidou et al., 2015*). Thus, mutations in Notch pathway participants that lead to defective signaling cause a variety of *human* diseases including neurodegenerative diseases, developmental disorders and cancer (Table 1).

## METHODS

### Dataset collection and filtering

Data was collected from the NCBI database (ncbi.nlm.nih.gov) as previously described in *Mitsis et al. (2020)*, towards to extracting the amino acid sequences that are related to the Notch proteins using the keyword "Notch" (*Mitsis et al., 2020*). Protein sequences that responded to the query but did not include the Notch family members were eliminated from the primary dataset, by using related keywords and regular expressions techniques in the header information, and local alignments with reference protein sequences. Furthermore, a final dataset for each species class was produced by using internal protein alignments and protein identity score. Duplicated protein sequences in each species that were found share 95% > protein identity within the dataset were removed. In total, 25,761 Notch family related protein sequences were identified from several species, and a dataset containing 603 unique, non-duplicate protein sequences was created (Dataset S1).

### Multiple sequence alignment

Multiple sequence alignment (MSA) was executed using the MATLAB Bioinformatics Toolbox, utilizing a guide tree and the progressive MSA method as previously described in several studies (*Mitsis et al., 2020*; *Papageorgiou et al., 2016*; *Sobie, 2011*). Pairwise distances among sequences were estimated based on the pairwise alignment with the

"Gonnet" method and followed by calculating the differences between each pair of sequences. The Neighbor-Joining method was used towards to estimating the guide tree by assuming equal variance and independence of evolutionary distance estimates (Dataset S2). Finally, consensus sequence was calculated and visualized through the JalView platform (*Waterhouse et al., 2009*) using the multiple sequences alignment results and parameters including amino acid conservation. The commentary section of Jalview, which presents the amino-acid conservation using logos and histograms, was further observed to uncover innovative motifs.

## Notch family protein clusters

Notch family protein clusters were identified by using phylogenetic analysis. The phylogenetic analysis was performed using the MATLAB Bioinformatics Toolbox (*Kufareva & Abagyan, 2012*) utilizing the Unweighted Pair-Group Method (UPGMA) (*Michener & Sokal, 1957*; *Pavlopoulos et al., 2010*; *Sneath & Sokal, 1973*) while the matrix of the pairwise distances was calculated using the protein-adapted Jukes-Cantor statistical method (*Papageorgiou et al., 2016*; *Yang & Zhang, 2008*). The constructed phylogenetic tree visualized using MEGA radiation option and the final Notch family protein clusters separated in different sub-datasets using a threshold. In total, eight major sub-clusters including Notch *Bacteria*, Notch_*Plants*, Notch_*Invertebrates*, Notch_*Protist*, Notch1, Notch2, Notch3 and Notch4 were identified (Dataset S3).

## Consensus sequences and a specialized phylogenetic analysis

Representative consensus protein sequences of the Notch family sub-clusters were estimated according to the clustering results of the Notch family protein clusters (*Mitsis et al., 2020*). The representative protein sequences for each sub-cluster were calculated using the MATLAB Bioinformatics Toolbox (*Nanni, Lumini & Brahnam, 2014*). Two representative protein sequences of the GLP1 and LIN12 homologs of the *Caenorhabditis elegans* species (*Girard et al., 2007*; *Sorkac et al., 2018*) were also included in the final dataset of the consensus protein sequences (Dataset S4). Last but not least, a specialized phylogenetic analysis was performed using the MATLAB Bioinformatics Toolbox (*Kufareva & Abagyan, 2012*) utilizing the Unweighted Pair-Group Method (UPGMA) (*Michener & Sokal, 1957*; *Pavlopoulos et al., 2010*; *Sneath & Sokal, 1973*) with 100 bootstrap replicates. Finally, the constructed topology of the phylogenetic tree was visualized with MEGA traditional option (Dataset S5).

## RESULTS

### Dataset

The primary NCBI dataset contained 26,646 entries, related to Notch family members. This dataset consisted of not only Notch protein and its homologs, but also unrelated proteins, such as the example of strawberry notch homolog (SNO). These irrelevant proteins, along with synthetic, hypothetical, partial, low quality, predicted proteins and which were referred as to noisy data and were removed from our dataset. Since the NCBI database provides partial duplicates sequences, the sequences with greater than 95%

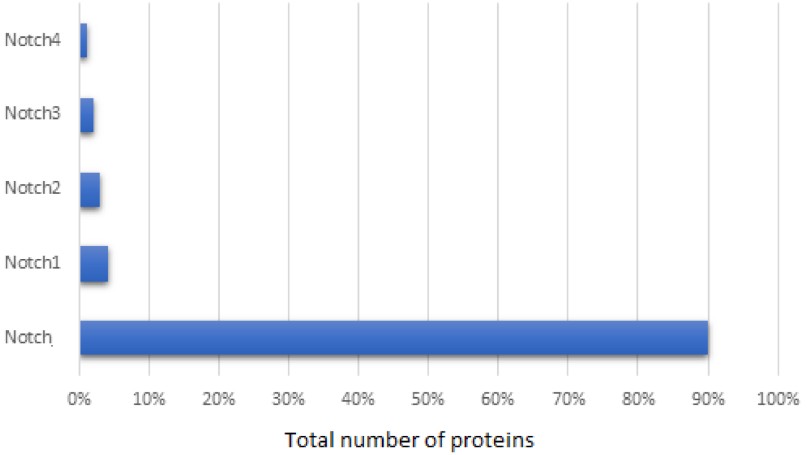

**Figure 1 Statistical analysis of the Notch family members based on the sequence annotation.**

similarity were also removed, retaining the one with the longer length. Thus, the final dataset involved 603 Notch proteins and specifically, 90% of these corresponded to Notch protein, 4% to Notch1, 3% to Notch2, 2% to Notch3, and 1% to Notch4 (Fig. 1). Herein, representatives of all kingdoms from simpler organisms such as *bacteria*, to more complex organisms including *Homo sapiens* were detected. The length of the Notch family proteins ranges between ≈71 aa (*bacteria*'s Notch-like protein) and ≈4,835 aa (*invertebrates* Notch). As for the Notch1–4 paralogues, an exponential pattern of length was observed, with N4 being the shortest, followed by Notch3, Notch2 and finally Notch1 being the longest.

## Multiple sequence alignment and motifs

Multiple sequence alignment (MSA) of protein sequences from the Notch family was performed to identify highly conservative regions within all organisms, from *monera* to *invertebrates*. As known, Notch receptors consist of specific conservative domains including EGF-like domain, NRR, TMD, RAM, NLS, ANK, NOD/NODP and PEST, that could be identified in the MSA (*Aster, Pear & Blacklow, 2017*; *Sander, Krysinska & Powell, 2006*) (Fig. 2). Monitoring and analyzing the MSA using Jalview resulted in identifying EGF, LNR and NOD/NODP domains appearance in most kingdoms, though a variation of the number of repeats and length is noticeable (*Gordon, Arnett & Blacklow, 2008*). Even *bacteria* Notch-like proteins share some of the characteristic motifs of those regions in their receptors; therefore, we can speculate that lateral gene transfer might have occurred via bacterial transfers (*Gazave et al., 2009*; *Ponting et al., 1999*). Two other members of the Notch pathway, including Fringe (*Irvine & Wieschaus, 1994)* and Strawberry Notch (Sno) (*Gazave et al., 2009*) show the same scenario, in which the eventuality of lateral gene transfer cannot be excluded.

Epidermal growth factors repeats are crucial components of the Notch signaling. Thus, it was expected to identify them as the most conserved site. EGF's 6 Cysteine residues, being the key element of the domain, are responsible for the formation of disulfide bonds

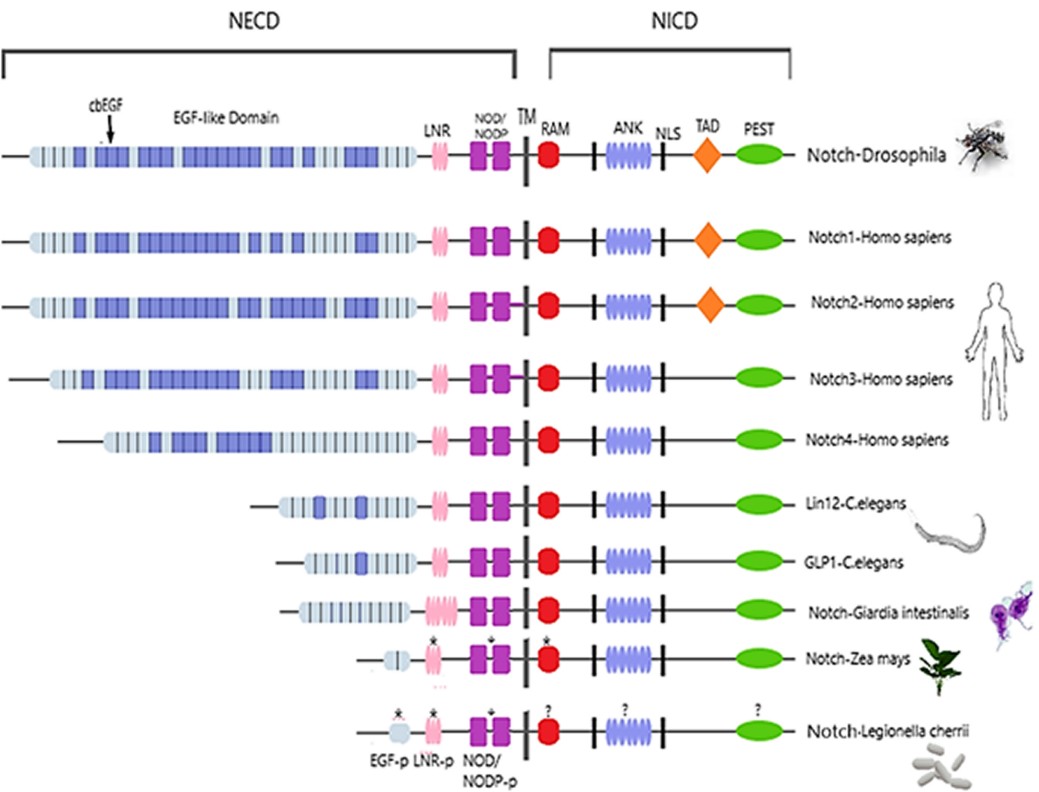

**Figure 2 Architecture of Notch family receptors based on the Interpro Database (*Mitchell et al., 2015*) results for all identified kingdoms.** Notch family receptors are represented with major domains annotated, including Notch extracellular domain (NECD), Notch intracellular domain (NICD), transmembrane domain (TM), Epidermal growth factor (EGF), Cysteine-rich LNR repeats (LNR), Notch domain present in many Notch proteins (NOD/NODP), RBPJκ-associated molecule domain (RAM), Nuclear localization sequences (NLS), Ankyrin repeat domain (ANK), and the Domain rich in proline, glutamine, serine and threonine residues (PEST). Protein domains marked with (*) represent domains that have only been observed in the MSA, and those marked with (?) represent domains for which no other information is available.

and influence the native 3D structure of the Notch members (Fig. 3). The importance of these key residues is also apparent in the pathological phenotype. In the case of a Notch3 mutation, if any of the 6 Cys is mutated into another amino acid, it leads to a rare neurodegenerative syndrome, called CADASIL (Cerebral Autosomal Dominant Arteriopathy with Subcortical Infarcts and Leukoencephalopathy) (*Papakonstantinou et al., 2019*; *Vlachakis et al., 2014*). It is worth mentioning that the EGF-domain is also characterized by the conserved Glycine residues, as shown by the current multiple alignments (Fig. 3). Furthermore, the highly conserved EGFs, led us to suggest that two new motifs A and B are crucial for the Notch family evolution. The CXNGGXC motif (Fig. 3/Motif A) consisting of two highly conserved Gly residues and a second motif (Fig. 3/Motif B), CXCXXG[FY]XG, characterized by a conserved Cys and Gly and a non-polar aromatic amino acid (F/Y) among two conserved Gly (Fig. 3). These motifs could possibly be εGF-domain's precursors, since they are repeated with a different

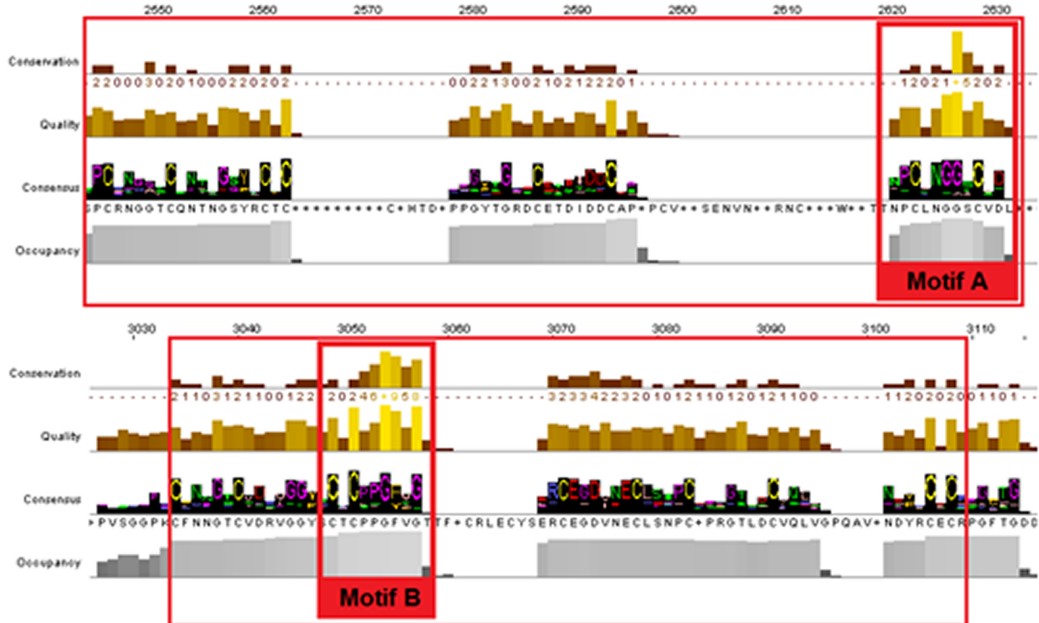

**Figure 3 Highly conserved Notch family motifs. Motif-A and Motif-B have been highlighted (colored red) in the EGF-domain, based on the Notch consensus sequence from the MSA.** The characteristic EGF-Cys-repeats are illustrated in with large red boxes.

number of repetitions depending on the respective species, where the number of repetitions is directly related to the complexity of the organism (*Kovall et al., 2017*).

Additionally, a segment of the LNR-domain appears to be highly conserved in organisms from all kingdoms and the characteristic motif of the LNR-repeat was also found in *bacteria*. Motif C could be considered as the LNRs' single unit of short peptide that has been repeated via internal tandem duplications through evolution in *eukaryotes* (*Bjorklund, Ekman & Elofsson, 2006*) (Fig. 4/Motif C). NOD/NODP domain seems to have maintained throughout the evolutionary history of all eukaryotes and it contains two conserved motifs (Motifs D and E). Remarkably, the conserved motif D has been identified also in *bacteria*'s Notch-like proteins, which are significantly smaller than the homolog proteins of the other species.

## Structural characterization of Notch Receptor

Despite the progress achieved recently, in the structure determination of limited fragments of the Notch receptor, its quaternary 3D structure remains undetermined (Fig. 5). Notch receptor 3D models are based on the structural features of the long NECD region, since it contains many calcium binding EGF-like domains (Fig. 5). The NECD differs between species. *Drosophila* and *mammalian* Notch receptors are much larger than their counterparts from other *invertabrate* species like *Caenorhabditis elegans*, although each invariably maintains the same molecular architecture (*Kopan & Ilagan, 2009*). On the other hand, in *monera* like *bacteria*, and in *protists*, the NECD size is much shorter and

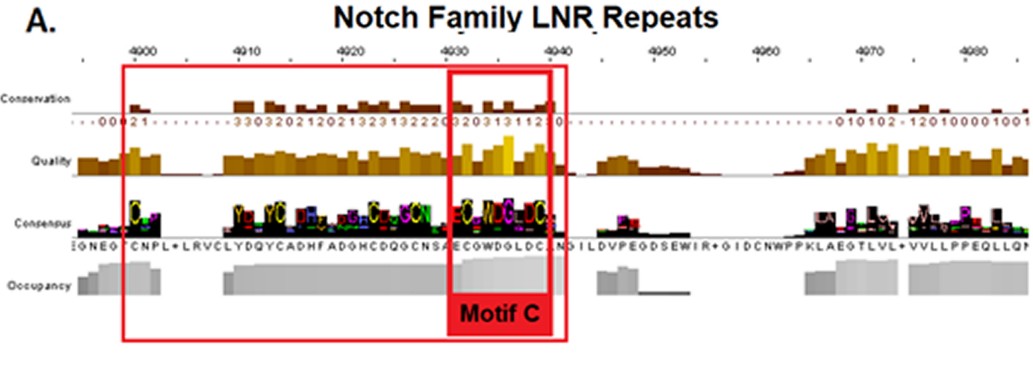

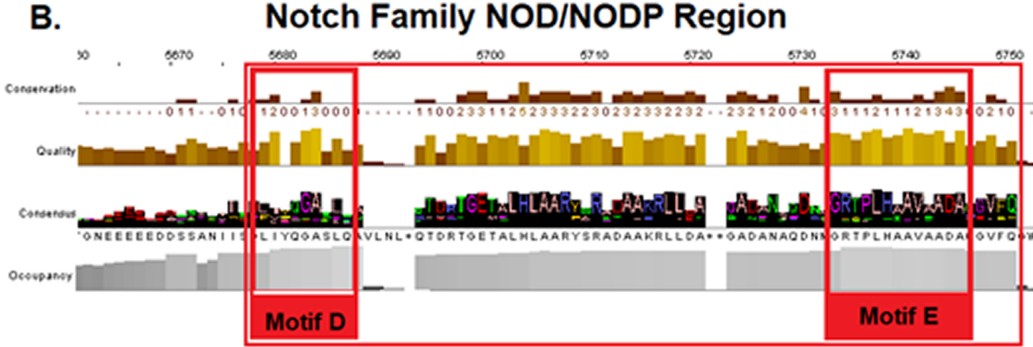

**Figure 4 Highly conserved Notch family regions based on the 603 Notch sequences MSA results.**
**(A) Conserved region of the LNR domain (B). Conserved regions of the NOD/NODP domain.**

compact (Fig. 2). Based on previous studies it has been observed that the NECD region of the Notch receptor is expected to have a rigid near-linear 3D structure, however potential sites of flexibility may occur at the 3D structure of the EGF domain which is less conserved between species/kingdoms (Figs. 2 and 3) (*Kopan & Ilagan, 2009*; *Morgan et al., 1999*).

At the N-terminal end, for most species, the Notch receptor contains 36 EGF-like domains, a region of containing calcium-binding sites. Next to the EGF region there are three LNR and a hydrophobic region which has been shown to mediate heterodimerization (HD) (Fig. 5). Together, the LNR repeats and the HD form the NRR, adjacent to the cell membrane (Fig. 5). This region prevents ligand-independent activation of the Notch receptor by concealing and protecting from metalloproteases. When Notch activation is achieved via ligand binding to repeats within the EGF-like domain, then two sequential proteolytic events termed S2 and S3 cleavages are induced (*Sanchez-Irizarry et al., 2004*). The S3 cleavage site lies within the transmembrane segment and is cleaved by the γ-secretase complex to liberate NICD. NICD comprises one RAM domain, seven ANK, one TAD and one PEST domain (Fig. 5). Both the RAM domain and ANK repeats have been identified as regions involved in the interaction with CSL transcription factors (*Chillakuri et al., 2012*). The TAD region is found in Notch-1 and -2 but not in -3 and -4 in *mammals* (*Chillakuri et al., 2012*). The C-terminal PEST domain is involved in NICD degradation by proteolysis.

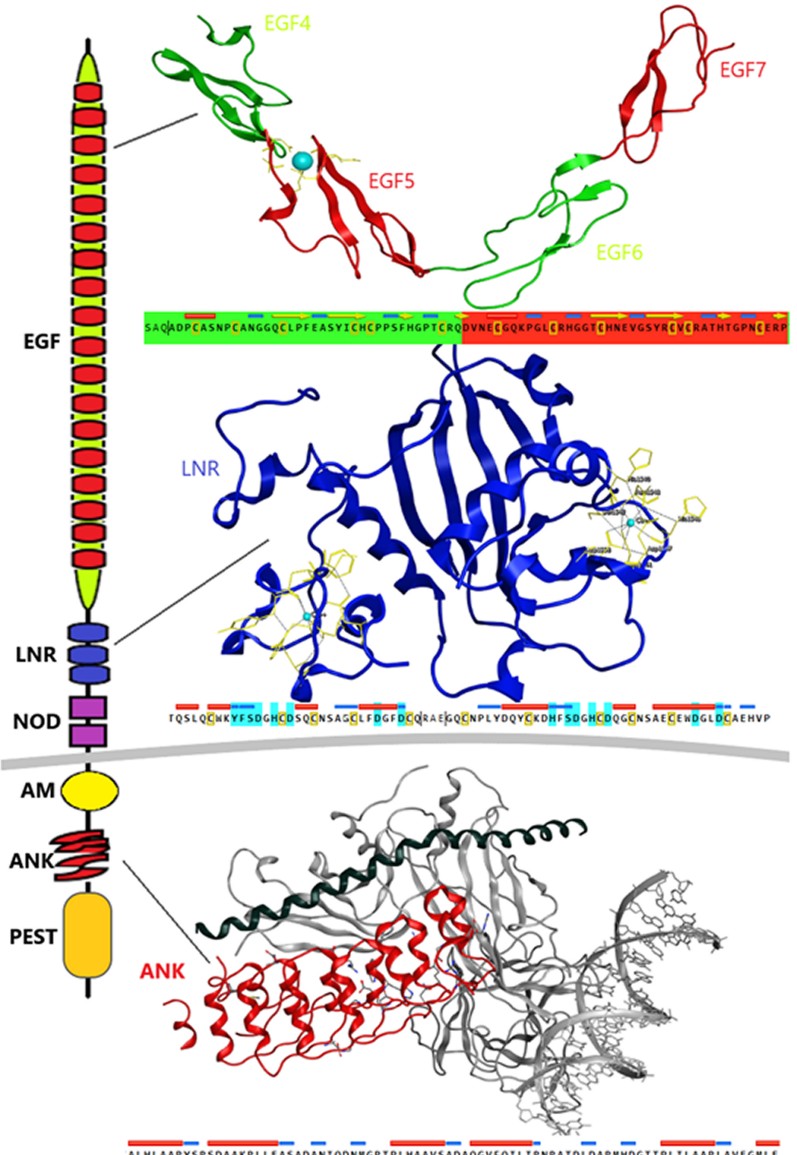

**Figure 5** Structural architecture of Notch family receptors based on the available fragments from the Protein Data Bank (PDB: 5FM9, 3ETO, 3V79).

Recently, they have been determined the complete 3D structures of the extracellular domain (ECD) of the *Drosophila* Notch receptor and the *human* Notch1 receptor, by using single particle electron microscopy and antibody labeling (*Chillakuri et al., 2012*; *Kelly et al., 2010*). Since the inherent resolution of the method is poorly corresponding to the small size of the domains involved, these are significant challenging experiments.

## Ca$^{+2}$ signaling activation and digital emulation

Notch family proteins, being capable of recognizing and binding to the neighbor cell's ligand, achieve selective cell-cell adhesion initiated by protein-protein interaction. They comprise an extracellular region, composed of EGF-repeats, LNR repeats and
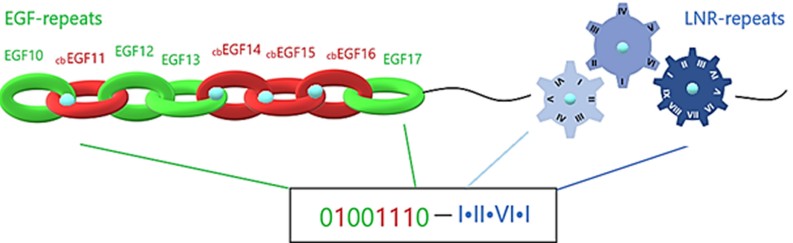

**Figure 6 Structural visualization of the ENCD of a Notch family receptor and its digital format.**

NOD/NODP. EGF-repeats act as transmitters that promote the signal to the LNRs and along those extracellular regions, intracellularly to the ANK repeats. In this process, calcium plays a crucial role and is required for the biological activation of transmembrane proteins that contain EGF and LNR repeats. More precisely, $Ca^{+2}$ binds to cbEGFs, a large subgroup of EGFs. CbEGFs have essentially similar structure as EGFs, but they are so conserved, that they cluster together in EGF dendrograms (*Stenflo, Stenberg & Muranyi, 2000*). The central role of $Ca^{+2}$ has been demonstrated through crystallization experiments, where $Ca^{+2}$ seems to stabilize the N-terminal of tandem EGFs, establishing a stable surface for the protein-protein interaction (*Downing et al., 1996*). The $Ca^{+2}$ binding coordination is related to cbEGFs' amino acid modifications, including hydroxylation and glycosylation of an asparagine or aspartic acid residue. Moreover, when a $Ca^{+2}$ is binding, the extracellular structure is locally changing, making it versatile and difficult to model. $Ca^{+2}$ role is also, observed in the LNR repeats, as each cysteine-rich Lin-12/Notch repeat binds one calcium ion to achieve the correct folding and maintenance of its 3-D structure. Mutations in calcium binding sites lead to dysregulated pathways, introducings pathogenic phenotypes. A substantial case of defective $Ca^{+2}$ binding is the Marfan syndrome and Hemophilia B.

The extracellular region of a Notch family precursor is a mix of calcium-loaded and calcium-free sites, with each EGF repeat forming a beta-sheet and the LNR repeats forming loops. Simulating each beta-sheet as a hoop, the whole EGF region could be presented as a strongly-connected chain. Furthermore, emulating the chain into a digital format, each hoop, depending on its $Ca^{+2}$ binding or $Ca^{+2}$ binding disability, is emulated by a sequence of 1 and 0, respectively. In that way, simulating each Lin12/Notch repeat as a gear (are more specialized forms than EGF repeats), the complex of LNRs could be emulated by a compact gear mechanism, which produces a more specialized signal based on the calcium interactions (Fig. 6). Combining all this mechanism together we can easily understand its specialization in producing different unique codes (signals) to the cell in order to start different biological tasks. This proposed emulation could represent a way to present if the signal is going to be promoted from the EGF repeats to the LNRs etc. and finally to the nucleus, completing a the signaling pathway. In case of mutation in these residues due to defective $Ca^{+2}$ binding, the corresponding digit changes and an error in the code occurs, indicating signal transduction failure associated with the possibility of a functional abnormality (Fig. 6).

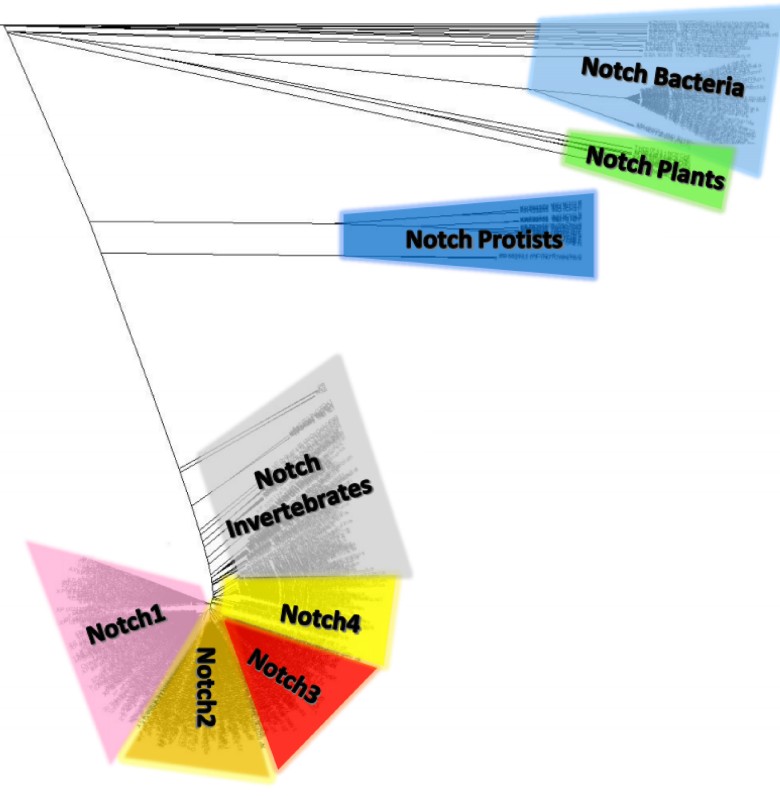

**Figure 7 The unrooted phylogenetic tree of Notch family members.** The tree was constructed utilizing the UPGMA method in MATLAB and visualized using the TreeExplorer tool of MEGA. Eight distinct monophyletic branches are visible. The phylogenetic trees confidently separate the Notch1 cluster (branch colored Pink), the Notch2 cluster (branch colored orange), the Notch 3 cluster (branch colored red), the Notch 4 cluster (branch colored yellow), the Notch Bacteria cluster (branch colored light blue), the Notch Plants cluster (branch colored green), the Notch Protist cluster (branch colored dark blue) and the Notch multicellular cluster (branch colored gray).

## Notch family protein clusters

Notch family protein clusters were estimated using a phylogenetic analysis (Fig. 7). A phylogenetic tree could provide clustering information for the Notch family members, so that representative consensus sequences could be properly defined. Having such a large dataset with proteins of variant sequence length, we initiated the construction of an unrooted tree. Phylogenetic analysis revealed a distinct separation of Notch family members into eight monophyletic branches, the Notch *bacteria* cluster, the *Notch* plants cluster, the Notch *protist* cluster, the Notch *invertebrates* cluster, as well as the Notch1, Notch2, Notch3 and Notch4 clusters. Each cluster's content was carefully examined and reviewed (proteins, classes and kingdoms it contained), eventually leading to clustering the data into the eight groups (Table 2). Moreover, it was clear that the resulting clusters of the phylogenetic tree were formed by related protein members and not by classes/ organisms, even if we included the same species in different clusters for the Notch1 to 4 homologs. Furthermore, considering the conservation and similarity of the characteristic domains presented in each Notch cluster, it was verified that the Notch family members

**Table 2 Categorization of the NOTCH family members using identified clusters, per kingdom and phylum.**

| Kingdoms | BACTERIA | Eukaryotes Single Cell PROTISTS | | | ANIMALS Invertebrates | | | | | Chordates | | | | | | PLANTS | FUNGI |
|---|---|---|---|---|---|---|---|---|---|---|---|---|---|---|---|---|---|
| Phylum | | Perkinsozoa | Apicomplexa | Flagellates | Placozoans | Cnidarians | Flatworms | Nematodes | Arthropods | Mammals | Fishes | Birds | Turtles | Amphibians | Lizards | | |
| NOTCH | ✓ 40 | ✓ | ✓ 8 | ✓ | ✓ | ✓ | ✓ 112 | ✓ | ✓ | – | – | – | – | – | – | ✓ 5 | ✓ 1 |
| NOTCH1 | – | – | – | – | ✓ | ✓ | ✓ | ✓ | ✓ | ✓ | ✓ 152 | ✓ | ✓ | ✓ | ✓ | – | – |
| NOTCH2 | – | – | – | – | ✓ | ✓ | – | ✓ | ✓ | ✓ | ✓ 114 | ✓ | ✓ | ✓ | – | – | ✓ 1 |
| NOTCH3 | – | – | – | – | ✓ | ✓ | ✓ | ✓ | ✓ | ✓ | ✓ 83 | – | ✓ | ✓ | ✓ | – | – |
| NOTCH4 | – | – | – | – | – | ✓ | – | ✓ | ✓ | ✓ | – 87 | – | ✓ | – | ✓ | – | – |

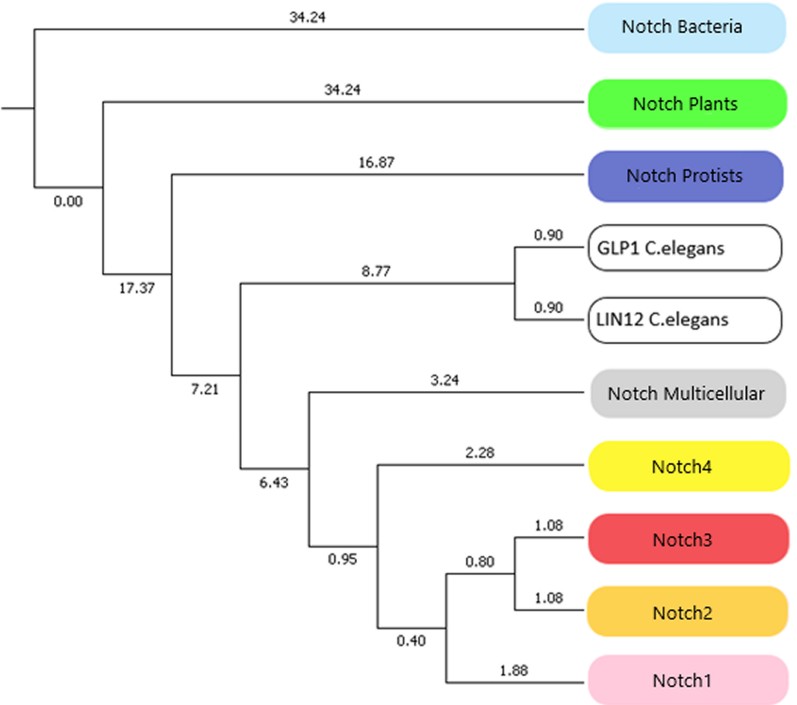

**Figure 8 The evolutionary history of Notch family members in a specialized phylogenetic tree.** The tree was constructed with UPGMA method in MATLAB for 100 bootstrap replicates and was visualized in MEGA.               

classification was accurate because they belong in the same phylum/kingdom (Fig. 7). In this study, using protein sequences from representatives of all kingdoms, it was necessary to observe the agreement of the emerging phylogenetic tree with the tree of life. Thus, a bibliographic examination was carried out, which confirmed the correct and proper positioning and clustering of organisms according to evolution, by examining the position of the *invertebrates*, such as cnidarians, flatworms, nematodes, arthropods, chordates, as well as the phylogenetic position of the protists and plants, in the tree of life (*Keeling, 2019*) (Table 2).

## Notch family evolution

In this study, we identified novel Notch protein clusters from various kingdoms extending from *bacteria* to *chordates* and conducted a comprehensive phylogenetic analysis in order to confirm and expand the evolutionary history of the Notch family (Fig. 8). Notch family evolution was examined in a more specialized phylogenetic analysis with 100 bootstrap replicates using representative consensus protein sequences of each identified cluster from the previous phylogenetic analysis. Each consensus sequence, with its amino acids corresponding to the most frequently encountered, fully represented the organisms and the members of the NOTCH family, accordingly. Hence, the specialized phylogenetic tree contained meaningful information, as needed for further evolutionary research (*Papageorgiou et al., 2018*). Based on results, both constructed phylogenetic trees of the Notch family members were found to share similar topology.

Due to the small number of sequences which were available the past decade, all previously performed evolutionary studies have a significantly smaller dataset. In this study, a more comprehensive phylogenetic analysis was achieved and it was important to compare the evolutionary scenarios presented here with the previous ones. The evolutionary distances between Notch2 and Notch3 groups have been found shorter than Notch, Notch1 and Notch4. This finding is correspondingly consistent with the evolutionary analysis performed by *Gazave et al. (2009)*. On the other hand, Notch4 group found to be evolutionary older than Notch1, Notch2 and Notch 3, and with this result being consistent with the evolutionary study performed by *Theodosiou et al. (2009)*. However, in this study we have introduced and present for the first time, the *plant*, *protist* and *bacteria* groups which are in the early Notch evolution (Fig. 8). Another remarkable observation is that Notch protein groups are positioned before Notch 1–4 groups, indicating the origin of the Notch family. In consideration of the tree of life, *bacteria*, *protists* and *plants*, appeared before the rest of the *eukaryotes* and therefore their position in the constructed tree is validated. *Bacteria* belongs to *prokaryotes*, while *protists* and plants are *eukaryotic* organisms and thus, much more complex than *prokaryotes*. The fact that the latter two are *eukaryotes*, combined with the absence of evidence for the existence of Notch proteins in *Archaea*, partially modifies the obtained results, compared to the current evolutionary theories. In addition, bearing in mind the limited number of representatives given in the kingdom of *plants*, this slight divergence in the present evolutionary study of the Notch family is justified. However, future enrichment of the dataset, with additional Notch family protein sequences from other species, could provide an improved and even more distinct phylogenetic analysis. Lastly, in the phylogenetic tree, Notch paralogs *of Caenorhabditis elegans* appear distinctly before Notch1–4, as expected, given the fact of an independent genomic duplication, leading to two Notch genes (*Theodosiou et al., 2009*).

## DISCUSSION

Central nervous system has evolved in early metazoans (such as marine organisms) as a simple neural network. This precursor system, necessary for cells' electrical signaling and cell-to-cell interaction, became more complicated through evolution (*Akanuma et al., 2002*). Notch family members form a key component of an evolutionarily conserved signaling mechanism, involved in the regulation of the CNS. Indeed, Notch signaling takes part in Neural Stem Cell (NSC) proliferation, survival, self-renewal, differentiation, apoptosis and cell fate choices (*Gazave et al., 2009*). In the CNS, Notch members are present during the entire lifetime, from embryonic stages to adult nervous system, controlling neurogenesis, axons' and dendrites' growth and CNS plasticity (*Presente, Andres & Nye, 2001*).

Several pathogenic mechanisms have been observed from different mutation cases in Notch receptor protein domains. Recent studies suggest that Notch receptors play correspondingly a crucial role in neurodegenerative disorders, including Alzheimer's disease, Down syndrome and CADASIL (*Polychronidou et al., 2015*). Notch family members are estimated that they existed a billion years before. Previous studies have

shown that Notch signaling in *Metazoan* is a subsequent evolutionary result, as it uses ancient protein domains and mechanisms. Thus, it was assumed that this pathway may exist in *Urmetazoa* or even in more inferior organisms, such as *bacteria* which also have Notch-like proteins (*Gazave et al., 2009*). Therefore, new insights have been extracted from the present in silico study, where it could form the basis for detecting unrelated functions in this receptor family. So far, the evolutionary history of the Notch family seems to be directly linked to the tree of life. In the same direction, the complexity of these proteins could be proportional to the complexity of the species and the nervous system, as well. Last but not least, *bacteria*'s undifferentiated Notch-like proteins seem to have been evolved and differentiated in all other types of paralogues Notch's as the complexity of the species increases. Similar evo-devo scenarios are claimed generally in proteins families, which are important for survival and evolution, such as the example of the nuclear receptor family (*Mitsis et al., 2019*).

## CONCLUSION

Despite the enormous scientific interest in Notch family proteins, current knowledge of their involvement in biological pathways and their function is still quite limited. Certainly, this knowledge could be significantly enriched with the potential determination of the Notch receptor 3D structure. Considering the cell-fate-determination and the cell communication, enforced by these proteins as well as the proteolytic procedure that they undergo in the signaling pathway, Notch receptors could be used as promising therapeutic targets for several diseases, including cancer. In the present study, a series of the most highly conserved motifs that have arisen through evolution is presented. These motifs could be used as innovative pharmacological targets, through the development of new technologies concerning the numerous critical pathways, affected by this protein family. All in all, useful beneficial insights are provided, concerning the Notch family's evolution. A comprehensive evolutionary analysis that confirms the existence of several kingdoms among Notch family members is provided. Notch genes were duplicated several times during evolution, leading to four genes in chordates, including *Homo sapiens*. If we accept that more than the half of our body is not *human*, then the most logical scenario for the ancient origin of Notch are the *bacteria*.

### Funding

Dimitrios Vlachakis received funding from: (i) Microsoft Azure for Genomics Research Grant (CRM:0740983); (ii) Amazon Web Services Cloud for Genomics Research Grant (309211522729); (iii) AdjustEBOVGP-Dx (RIA2018EF-2081): Biochemical Adjustments of native EBOV Glycoprotein in Patient Sample to Unmask target Epitopes for Rapid Diagnostic Testing. A European and Developing Countries Clinical Trials Partnership (EDCTP2) under the Horizon 2020 "Research and Innovation Actions" DESCA. Elias Eliopoulos received funding by the project "INSPIRED-The National Research Infrastructures on Integrated Structural Biology, Drug Screening Efforts and Drug Target

Functional Characterization" (Grant MIS 5002550) and by the project: "OPENSCREEN-GR An Open-Access Research Infrastructure of Chemical Biology and Target-Based Screening Technologies for Human and Animal Health, Agriculture and the Environment" (Grant MIS 5002691), which are implemented under the Action "Reinforcement of the Research and Innovation Infrastructure", funded by the Operational Program "Competitiveness, Entrepreneurship and Innovation" (NSRF 2014-2020) and co-financed by Greece and the European Union (European Regional Development Fund). The funders had no role in study design, data collection and analysis, decision to publish, or preparation of the manuscript.

## Grant Disclosures
The following grant information was disclosed by the authors:
Microsoft Azure for Genomics Research Grant: CRM:0740983.
Amazon Web Services Cloud for Genomics Research Grant: 309211522729.
AdjustEBOVGP-Dx: RIA2018EF-2081.
A European and Developing Countries Clinical Trials Partnership (EDCTP2).
Greece and the European Union (European Regional Development Fund): MIS 5002550, MIS 5002691 and NSRF 2014-2020.

## Competing Interests
The authors declare that they have no competing interests.

## Author Contributions
- Dimitrios Vlachakis conceived and designed the experiments, performed the experiments, analyzed the data, prepared figures and/or tables, authored or reviewed drafts of the paper, and approved the final draft.
- Louis Papageorgiou performed the experiments, analyzed the data, prepared figures and/or tables, authored or reviewed drafts of the paper, and approved the final draft.
- Ariadne Papadaki performed the experiments, analyzed the data, authored or reviewed drafts of the paper, and approved the final draft.
- Maria Georga performed the experiments, analyzed the data, authored or reviewed drafts of the paper, and approved the final draft.
- Sofia Kossida conceived and designed the experiments, performed the experiments, analyzed the data, authored or reviewed drafts of the paper, and approved the final draft.
- Elias Eliopoulos conceived and designed the experiments, performed the experiments, analyzed the data, authored or reviewed drafts of the paper, and approved the final draft.

## Data Availability
The raw evolutionary, datasets and modeling measurements are available in the Supplemental Files.

## Supplemental Information
Supplemental information for this article can be found online at http://dx.doi.org/10.7717/peerj.10334#supplemental-information.

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
