# Peer review of "An updated evolutionary study of the Notch family reveals a new ancient origin and novel invariable motifs as potential pharmacological targets"

_PeerJ, doi:10.7717/peerj.10334_

## Round 0.1 · original submission · Major Revisions

Please address critiques of all reviewers, paying careful attention to comments of reviewer #2.

·

Basic reporting

No comment

Experimental design

No comment

Validity of the findings

No comment

Additional comments

In this manuscript, the authors searched and analyzed Notch protein sequences from organisms of all kingdoms. As a result, a large number of Notch proteins are identified in bacteria, protists, fungi, plants and invertebrates in addition to the previously well characterized Notch1~4 proteins. Based on evolutionary analysis to 603 unique Notch protein sequences, they inferred existence of ancestral Notch sequences in bacteria, protists and plants. Furthermore, they successfully identified characteristic motifs A to E in EGF domain. These motifs have conserved glycine(s) in addition to conserved cysteines, which can be potential pharmacological targets for future studies. The results present in this paper enrich our knowledge on distribution, evolution and structure of Notch proteins in various organisms. I would recommend publication of it in PeerJ after the following minor revisions are made.

Lines 39 to 48: The abstract has too many words as ‘introduction’. Please reduce them and add words to describe results and significance of your work.

Line 109: Please specify the query item. Is it the word “Notch”?

Line 161: The number is 613 here. It is 603 at other places (line 116 etc). Also, line 49 says “603 different organisms”. But, line 116 says “603 unique, non-duplicate protein sequences”. Please check.

Lines 223 to 236: (Figure 5) should be (Figure 6), I guess.

Line 278: (Figure 6) should be (Figure 7), I guess.

Line 452: The reference is a duplicate of line 450.

Table 2: It is better to replace the tick with number of Notch genes found in each phylum/class among the 603 unique sequences. Also in Table 2 (and in the whole manuscript), it is better to replace “Multicellular” with “Invertebrates”, because multicellular organisms include chordates.

Figure 3: Please specify how many sequences are used to generate this graph. Is it the same with Figure 4 (603 Notch sequences)?

Figure 7: It is better to put the color block beneath tree branches but not on them, so that the tree branches can be visualized. Besides, please indicate how many sequences were used for construction of this tree.

Reviewer 2 ·

Basic reporting

In this paper, the authors aims to provide an updated evolutionary history of the Notch family members. This question is very interesting as the last paper dealing with it may be a bit old from now and new genomic data coming from various species may help to better understand and dissect the evolution of this family. However, the paper in this state does not allow to answer this question, due to several flaws in the methodology used (see below). Also the evolutionary considerations are not accurate (ex : l63 "indicating the progressive differentiation of Notch family"; l163 "simpler organisms", l349 "inferior organisms", L335 "the CNS has evolved in early metazoans, referring to ascidians which are far from early metazoans etc etc; ;phylogeny "map" (abstract), separation of the animals into "multicellular vs chordates" in the table 2 etc etc... . I am also not sure the literature references provided are relevant and accurate, for ex in the introduction the authors cites "Durieux et al 2019" and" Fairclough et al 2013" to describe the length of the notch proteins in bacteria and animals respectively. I did not see any mention of Notch protein in those papers and I do not think that notch proteins do exist in bacteria (maybe a specific domain of the notch protein, but not the protein itself). Finally, the link with pharmacological targets is not clear to me.

Experimental design

The most important issue of the paper is the selection of the proteins to be included in the analyses. The way to select/identify proteins in order to do comparative genomics is to blast a protein query (here probably human) against databases of interest. The key word search is not an alternative way of doing it, as it is biaised (it depends on the way the protein has been submitted to the database) and highly incomplete. 2nd, the authors needs to define what is the minimal domain composition of a protein to be considered as notch. The presence of one or few domains, not specific to a notch proteins is probably not sufficient. Short "notch" sequences from bacteria such as Legionella X, have even no domain detected (search by NCBI CD search tool). How the authors deal with such different sizes of proteins ? how the alignment for the tree was done ? What about the missing data ? Did the author select just specific domains ? what about the variable number of EGF ? The notch protein phylogeny is quite hard due to all this specificities. Surprisingly, the authors used a very old method, not used any more to do their phylogenetic reconstructions (UPGMA), with no or few method of statistical support.

Validity of the findings

Based on the flaws mentioned above, the results provided are not conclusive.
The tree (figure 7) is non resolved. In the figure8, what are the number on the branches ?
Why the authors made a separation between multicellular and chordates ? are chordates unicellular ?

·

Basic reporting

Authors has made excellent efforts in compiling the MS with detailed, clearly, consistently communicated a rational design followed by their outcomes, interpretation and significance of the work.

Experimental design

No changes required in the compiled version

Validity of the findings

Author has made great effort in contributing the new novelty in the present MS. My detail comments are in the author section.

Additional comments

The topic seems excellent addition in the current literature however, before accepting for the publication, it should advisable to refine further by making necessary minor corrections as suggested in the bottom which should provide more clarity to readers.

Comments on Manuscript
Couple of Comments/suggestion should be incorporate to improve English, Formatting and Uniformity.
1. Title-Please include the before the Notch family
2. Affiliation 5, cedex should have ‘C’ in capital letter
3. Line 50, abstract, please add ‘s’ after human
4. Line 73, Introduction: replace ‘to’ with ‘in’
5. Line 76, provide symbol for approximation and replace the word amino acids with ‘aa’ for uniformity
6. Line 97, insert comma after TAD domain
7. Line 97, remove the word in before the length
8. Line 98, remove the before vascular…..
9. Line 101, add ‘the’ after involved in, so sentence looks like The pathway is involved in the….
10. Line 102, Insert comma, after learning
11. Line 113, insert ‘s’ after score; 114, replace which with that; Line 115, symbol will come before 95%
12. Line 137, insert ‘a’ before threshold
13. Line 158, insert ‘to’ before as noisy….
14. Line 136, in the Figure 1, author should provide axis information along with as an insert total number of protein
15. Line 166, insert amino acids or aa after 4835
16. Line 185, replace a with the before most conserved site…..
17. Line 193, insert ‘s’ after alignment…
18. Line 196, delete ‘a’ before non-polar
19. Line 198, add they after since….are repeated….
20. Line 205, insert ‘the’ before evolutionary history……
21. Line 206, replace the It’s remarkable that with Remarkably,……
22. Line 207, insert ‘ly’ after significant
23. Line 217, Correct the spelling of Caenorabditis……
24. Line 221, insert ‘s’ after appear
25. Line 224, delete ‘of’ before containing calcium…..
26. Line 225 and 226, replace ‘s’ with ‘z’ for the word heterodimeri….
27. Line 237, insert ‘the’ before human notch1
28. Line 243, charge on calcium should be superscript and similar mistakes should be corrected other places
29. Line 275, add with and delete to comes before the possibility,,,,,
30. Line 320, rewrite the sentence like be at the beginning of…
31. Line 343, remove ‘A number of’ and start the statement with Several……., replace has with have
32. Line 347, insert ‘s’ after year… and replace showed with shown….
33. Line 374, remove ‘the’ before half of our…..
34. Line 375, remove are before the bacteria..
35. Line 450-452; 472-475; 530-533, remove the additional duplicate reference from the MS.

---

## Round 0.2 · accepted · Accept

Since all the critiques of the reviewers were adequately addressed, and the manuscript was revised accordingly, I am pleased to accept your manuscript for publication.

·

Basic reporting

Fine

Experimental design

Fine

Validity of the findings

Fine

Additional comments

All my concerns on the original manuscript have been properly addressed by the authors. I am comfortable with the revised paper.

·

Basic reporting

Authors is advised to proof read again the MS as there are very minor improvement is required in terms of grammar.

Experimental design

Authors has made necessary revisions in the revised MS

Validity of the findings

No Changes

Additional comments

The authors has made suggested changes in the revised MS and I do not have any additional correction in the present form of MS